# Developing and evaluating a brief, socially primed video intervention to enable bystander cardiopulmonary resuscitation: A randomised control trial

Jean Skelton[1,2]*, Anne Templeton[2], Jennifer Dang Guay[2], Lisa MacInnes[1], Gareth Clegg[1,3]

1 Usher Institute, University of Edinburgh, Edinburgh, Scotland, United Kingdom, 2 School of Philosophy, Psychology, & Language Sciences, University of Edinburgh, Edinburgh, Scotland, United Kingdom, 3 Scottish Ambulance Service, Edinburgh, Scotland, United Kingdom

* jean.skelton@ed.ac.uk

**Data Availability Statement:** All relevant data are publicly available on the Open Science Framework

## Abstract

### Background

Over 30,000 people experience out-of-hospital cardiac arrest in the United Kingdom annually, with only 7–8% of patients surviving. One of the most effective methods of improving survival outcomes is bystander intervention in the form of calling the emergency services and initiating chest compressions. Additionally, the public must feel empowered to act and use this knowledge in an emergency. This study aimed to evaluate an ultra-brief CPR familiarisation video that uses empowering social priming language to frame CPR as a norm in Scotland.

### Methods

In a randomised control trial, participants ($n = 86$) were assigned to view an ultra-brief CPR video intervention or a traditional long-form CPR video intervention. Following completion of a pre-intervention questionnaire examining demographic variables and prior CPR knowledge, participants completed an emergency services-led resuscitation simulation in a portable simulation suite using a CPR manikin that measures resuscitation quality. Participants then completed questionnaires examining social identity and attitudes towards performing CPR.

### Results

During the simulated resuscitation, the ultra-brief intervention group's cumulative time spent performing chest compressions was significantly higher than that observed in the long-form intervention group. The long-form intervention group's average compressions per minute rate was significantly higher than the ultra-brief intervention group, however both scores fell within a clinically acceptable range. No other differences were observed in CPR quality. Regarding the social identity measures, participants in the ultra-brief condition had greater

(https://osf.io/feh54/; DOI 10.17605/OSF.IO/FEH54). This link is also listed within the manuscript.

**Funding:** This project was made possible by a Research & Development Grant from Resuscitation Council UK (www.resus.org.uk). GC was lead applicant on the grant application, and JS received salary payment from the grant as lead research assistant. Grant number: 2020-4105393130. The funders had no role in study design, data collection and analysis, decision to publish, or preparation of the manuscript.

**Competing interests:** The authors have declared that no competing interests exist.

feelings of expected emergency support from other Scottish people when compared to long-form intervention participants. There were no significant group differences in attitudes towards performing CPR.

## Conclusions

Socially primed, ultra-brief CPR interventions hold promise as a method of equipping the public with basic resuscitation skills and empowering the viewer to intervene in an emergency. These interventions may be an effective avenue for equipping at-risk groups with resuscitation skills and for supplementing traditional resuscitation training.

## Background

Each year over 30,000 people in the United Kingdom (UK) experience out-of-hospital cardiac arrest (OHCA), but only 7–8% of patients survive to hospital discharge [1,2]. The most effective method for improving survival from OHCA is prompt bystander action in the form of cardiopulmonary resuscitation (CPR) [3,4], which has been shown to double or even triple OHCA survival outcomes [5]. In order to facilitate bystander intervention, members of the public need to be equipped with CPR knowledge and feel empowered to use their CPR skills in an emergency [6].

Current methods of CPR training are generally long, information-heavy, and delivered by perceived experts [7,8]. They require a significant investment of time and attention and can give the impression that CPR is complicated and should be performed by skilled professionals [9,10]. Furthermore, those most likely to be a bystander in an OHCA are often those not captured by traditional CPR training, such as people from areas of multiple deprivation and older people [11,12].

One approach to reducing these barriers to learning CPR is to use an 'ultra-brief video' (UBV) CPR intervention. A UBV CPR intervention is a short, shareable method of enabling the viewer to identify a cardiac arrest while equipping them with a fundamental knowledge of compression-only CPR. Participants exposed to CPR UBVs are more likely to attempt CPR, have higher quality CPR, and have higher self-reported confidence in their CPR abilities in comparison to untrained laypersons [10,13,14], emphasising the potential utility of this methodology in widening access to CPR familiarisation.

While CPR knowledge is a key component to improving OHCA survival, it is not the only barrier to prompt bystander action [9]. A survey conducted in Scotland found that only 72% of trained individuals felt confident to use their CPR skills in an emergency [15]. Similarly, a UK-wide survey of 4,000 participants found that only 51% of respondents felt confident to perform CPR in an OHCA [16]. One potential approach to increasing willingness to perform bystander CPR is to emphasise the role of social identity in emergency contexts. Social identity refers to people's memberships of particular social groups, such as their occupation or nationality. These social groups have associated social norms, which dictate the rules and beliefs of what it means to be a part of that group. Social norms guide behaviour and can be pivotal for the provision of social support [17–19]. For example, demonstrating that helping others is an integral part of a national group's identity has led to widespread helping behaviours at the country level [20]. Notably, having a shared social identity (i.e., the perception that people feel part of the same group) is important in emergency contexts, as people are more likely to coordinate with others and trust their information when they feel part of the same group [21,22].

In light of these findings, this study aimed to create and evaluate a maximally accessible UBV CPR intervention that frames CPR as a social norm in Scotland. In order to highlight CPR as a norm, collectivist language was primed in the script, using phrases such as "In Scotland, we look out for each other". These calculated language choices were designed to increase the salience of Scottish identity, empowering the viewer to see bystander CPR as a norm in Scotland and, by extension, to use their CPR skills when needed.

While UBVs have been shown to facilitate better CPR quality than control conditions, less is known about their efficacy in comparison to traditional, long-form CPR training videos (LFVs). In the current study, a randomised control trial was designed to compare the proposed UBV with the 2020 16-minute British Heart Foundation (BHF) 'Call Push Rescue' compression-only CPR familiarisation video [23]. The Call Push Rescue video was chosen for this comparison as it is a commonly used online training tool in the UK. In addition to the familiarisation video, each intervention included a practical CPR component to enable compression practice, an important quality predictor in CPR familiarisation [14]. For the LFV condition, the BHF 'Call Push Rescue' video incorporates the use of a Laerdal Mini Anne manikin, therefore LFV participants used this manikin for the associated compression practice. As the UBV had no specific associated manikin type, Save a Life for Scotland (SALFS) CPR bags were used for compression practice as these bag manikins are freely available to the Scottish public, corresponding with the accessibility ethos of the UBV intervention (see Fig 1 for examples of each manikin). Therefore, the interventions being testing are the CPR familiarisation videos in conjunction with the associated manikin, as opposed to just the familiarisation video in isolation.

Participants then took part in a simulated OHCA scenario in which they performed bystander CPR on a Laerdal Little Anne Quality CPR (QCPR) manikin, which measures multiple CPR quality metrics using wireless sensors embedded in the manikin. In order to maximise ecological validity, the simulations included a mobile phone which connected to trained research staff acting as emergency services call handlers to guide the participants in the basics

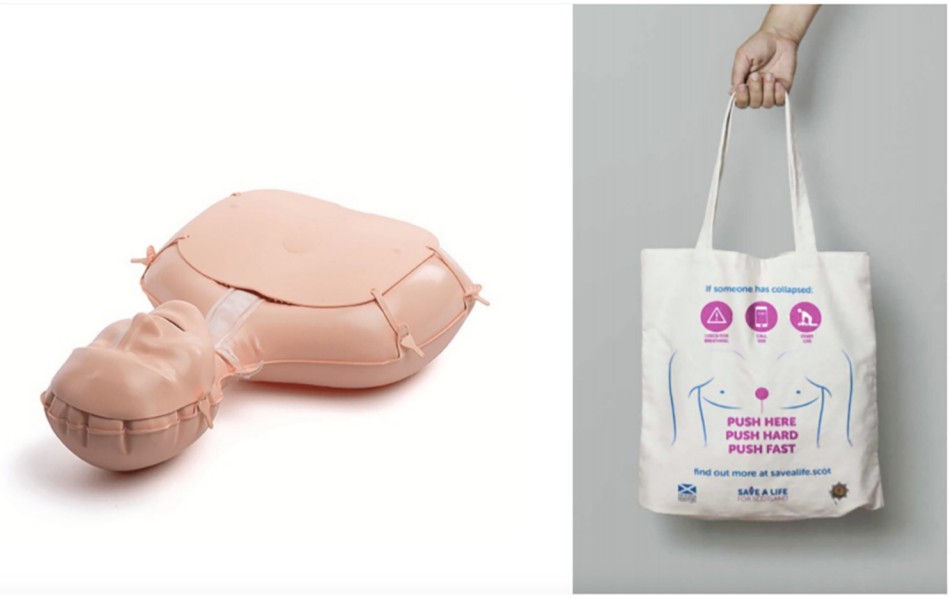

**Fig 1. Materials used for compression practice.** Left side: British Heart Foundation Call Push Rescue inflatable Mini Anne manikin. Right side: Save a Life for Scotland CPR bag. By filling the CPR bag with clothing or a pillow, the user can practice chest compressions.

of bystander resuscitation, as is standard in real OHCA emergency services calls in the UK [24]. Simulations were filmed using a multi-camera, portable Motorola simulation suite to generate further data on the accuracy and validity of the simulations.

*Study aims:*

1. Develop a compression-only CPR familiarisation intervention video which is short, shareable, and employs a social identity approach with the goal of maximising accessibility to CPR knowledge and increasing the likelihood of performing effective CPR.

2. Evaluate the effectiveness of this UBV intervention in comparison to a longer-form, traditional CPR familiarisation intervention in relation to emergency services-led CPR quality.

3. Examine group differences between the UBV and LFV conditions with respect to social identity, shared social identity with Scottish people, expected support from Scottish people, trust in the video instructor, and perceiving CPR as a social norm in Scotland.

4. Examine group differences between the UBV and LFV conditions in terms of willingness to perform CPR, confidence in ability to perform CPR, perceived barriers to CPR initiation, and understanding of the respective CPR familiarisation videos.

*Hypotheses:*

1. There will be no statistically significant differences between the UBV and LFV groups in terms of CPR quality metrics as measured by the QCPR manikin.

2. The UBV group will score higher than the LFV group on measures of social identity, indicating greater social identification with the UBV video instructor and other Scottish people.

3. There will be no statistically significant differences between the UBV and LFV groups on the variables of willingness to perform CPR, confidence in ability to perform CPR, perceived barriers to CPR initiation, and understanding of the respective CPR familiarisation videos.

*Primary outcome variable:*

1. CPR quality as measured by the QCPR manikin.

   Secondary outcome variables:

1. Social identity as measured by the post-intervention questionnaire.

2. Attitudes towards the assigned video as measured by the post-intervention questionnaire.

3. Attitudes towards performing CPR as measured by the post-intervention questionnaire.

## Methods

### Design

Two prior between-subjects experimental sub-studies were conducted to develop and test the socially primed UBV script. Further details on these studies can be found in the Supplementary Materials (S1 Appendix). Once the script was finalised, a test video was created. A parallel randomised control trial design with a one-to-one allocation ratio was employed. Participants were randomly assigned to one of two conditions: the refined 2-minute UBV, or a traditional, longer-form BHF CPR familiarisation video. A priori sample size calculation using G*Power based on the large effect sizes observed in previous literature [10,14] yielded a required sample

**Table 1. Descriptive statistics for demographic variables in each condition.**

| Variable | UBV (*n* = 41) | % | LFV (*n* = 40) | % |
|---|---|---|---|---|
| **Age Range** | 18–24 | 71 | 18–24 | 50 |
| | 25–34 | 22 | 25–34 | 30 |
| | 35–44 | 7 | 35–44 | 10 |
| | | | 45–54 | 5 |
| | | | 55–64 | 5 |
| **Gender** | Female | 63 | Female | 57.5 |
| | Male | 15 | Male | 22.5 |
| | Non-binary | 2 | Non-binary | 2.5 |
| | Prefer not to say | 20 | Prefer not to say | 17.5 |
| **Region of Residence** | Scotland | 83 | Scotland | 85 |
| | Outside UK | 5 | Outside UK | 2.5 |
| | Not Disclosed | 12 | Northern Ireland | 2.5 |
| | | | Not Disclosed | 10 |
| **Employment** | Employed full time | 9.8 | Employed full time | 22.5 |
| | Employed part time | 2.4 | Employed part time | 5 |
| | Student | 85.4 | Student | 70 |
| | Unemployed | 2.4 | Unemployed | 2.5 |

A summary of the descriptive statistics for the demographic variables. Percentage of the total sample per condition is displayed. No significant differences were found between groups on the basis of demographic variables (p > 0.05).

size of 74 participants (80% power, alpha = 0.05, two-sided test) to detect a significant difference in the primary outcome variable of CPR quality between groups.

## Participants

Participants (*n* = 86) were recruited from staff and students at the University of Edinburgh. Descriptive statistics for the demographic variables are shown in Table 1. Exclusion criteria were being under 18 years of age, being a healthcare professional, being CPR trained in the last 18 months, and not being a staff member or student at the University of Edinburgh. Four participants were excluded for not meeting these criteria, resulting in a sample of 82 participants.

In relation to the QCPR data specifically, seven participants were excluded due to technical issues with the QCPR manikin or ending the simulation early due to exertion (UBV: *n* = 4, LFV: *n* = 3), resulting in a final QCPR sample size of 75 participants (UBV *n* = 38, LFV *n* = 37).

Regarding the pre-intervention questionnaire, complete data for 81 participants were collected (UBV *n* = 41, LFV *n* = 40). A single participant's data was lost due to technical issues. For the post-intervention questionnaire, complete data were collected for all 82 participants (UBV condition *n* = 42, LFV condition *n* = 40). A flowchart of participant recruitment and exclusion can be seen in Fig 2.

## Materials

Once written consent was obtained, participants completed a pre-intervention questionnaire, which included information about demographic characteristics and prior CPR knowledge, and a post-intervention questionnaire, which examined social identity, feedback on the assigned video intervention, and attitudes towards performing CPR. These measures were

**Brief CPR Video Testing**

**Fig 2. Study protocol.** Flow chart detailing the recruitment and testing process of the randomised control trial.

validated during the UBV video development (see S1 Appendix). The pre-intervention demographic variables were: age range, gender, region of residence, employment, and prior CPR knowledge. Participants also completed a screening measure at this time to ensure they met the inclusion criteria.

Regarding the CPR interventions, the BHF 'Call Push Rescue' video and associated Mini Anne manikin were used in the LFV condition, while the novel SALFS 2-minute CPR

familiarisation video and SALFS bag manikin was used as the UBV intervention. QCPR metrics were recorded using the Laerdal QCPR App (Version 5.1.0; 25). The following CPR quality metrics were examined: compression depth, which measures percentage time spent performing compressions at the correct depth; chest compression fraction (CCF), which measures percentage time where adequate compressions are occurring; release of compressions, which quantifies adequate release of a compression; total number of compressions; percentage of compressions between the optimum rate of 100–120 beats per minute (bpm); and average number of compressions per minute. An overall compression score and total composite CPR score were also examined, as calculated by a scoring algorithm developed by Laerdal Medical. See S2 Appendix for further information on the calculation and operationalisation of the QCPR metrics. All simulations were filmed using Motorola Solutions VB400 cameras in a portable, multi-angle simulation suite. Motorola VideoManager software (Version 16.2; 26) was employed in order to gather feedback from Scottish Ambulance Service paramedics and call handlers on the quality and ecological validity of the simulations.

The post-questionnaire variables examined in relation to social identity were: Scottish social identity, or the degree to which participants identify with Scottishness; shared Social identity with the video instructor, or the extent to which participants identified as being in the same social group as their video instructor; trust in the person providing instruction, which examined participant trust in the video instructor; perceived norm of helping, or the extent to which participants associated helping behaviours with Scottishness; and expected support, which examined expected levels of support from Scottish people in an emergency. Social identity questionnaire items were scored on a Likert scale ranging from 1 ("Strongly Disagree") to 5 ("Strongly Agree").

With regard to attitudes towards the assigned video intervention, the post-intervention questionnaire captured participant views on the following variables: anxiety from the video instructions, which examined participants' anxiety towards the video instructor; sufficiency of practical CPR information, or the extent to which participants felt their video furnished them with sufficient information; and clarity, which examined how clear the participants found their video. Regarding attitudes towards performing CPR, the post-intervention questionnaire examined confidence in performing CPR, which examined participants' confidence in their CPR ability; willingness to perform CPR, which explored participant willingness to perform bystander CPR; and barriers to CPR performance, which explored participants' perceived barriers to performing bystander CPR. Questionnaire items that explored attitudes towards performing CPR and the assigned video intervention were scored on a Likert scale ranging from 1 ("Strongly Disagree") to 5 ("Strongly Agree"). All materials and data used are available on the Open Science Framework [27].

## Procedure

Ethical approval was secured from the University of Edinburgh's Medical Education Ethics Committee (Reference: 2022/02). Participants were randomised to their assigned condition prior to participation using an algorithm-based computer randomisation programme, "Research Randomizer" [28]. The randomisation sequence was stored on an encrypted file on OneDrive. Participants, but not research staff, were blinded to their condition. All allocation and recruitment was carried out by the lead research assistant from the 1st of July 2022 to the 28th of February 2023 in the Queen's Medical Research Institute, University of Edinburgh.

On arrival, participants completed the pre-intervention questionnaire using a digital form on a study laptop. Participants then watched their assigned video and completed the associated compression practice before undergoing a two-minute simulation of an OHCA in the

simulation suite. At the beginning of the simulation, participants were given a smart phone to call a preloaded emergency services contact. This call went to a research assistant trained in using the Medical Priority Dispatch System (MPDS), the emergency call handling system used for dispatching the emergency services in the UK and guiding bystanders through the fundamentals of CPR. Scottish Ambulance Service call handlers trained the research team in the proper use of this script and reviewed the simulation videos for quality control. During the simulation, participants were required to correctly identify a cardiac arrest, call the emergency services, and perform 2-minutes of emergency services-led bystander CPR on the QCPR manikin. After 2-minutes of bystander CPR, the research assistant terminated the simulation. Participants then completed the post-intervention questionnaire using a digital form on a study laptop. A detailed overview of the study protocol is available on the Open Science Framework [27].

## Analysis

Analyses were conducted using R version 4.2.2 [29] and R Studio version 2023.03.1+446 [30]. Shapiro-Wilk tests were used to assess normality of the data for each outcome variable, which were not normally distributed ($p < 0.05$). Given the small sample size and the categorical nature of the demographic data, Fisher's Exact Test of Independence was used to examine significant inter-group differences in demographic variables. Wilcoxon-Mann-Whitney tests for independent samples were used to examine average differences between the UBV and LFV groups for the variables of prior CPR knowledge, QCPR metrics, social identity, attitudes towards the assigned video intervention, and attitudes towards performing CPR. Benjamini-Hochberg procedures were used to adjust for multiple comparisons. Wilcoxon-Mann-Whitney tests for independent samples were reported using medians (Mdn), interquartile ranges (IQR), and standardised $Z$-scores, while effect sizes were reported as rank-biserial correlation coefficients ($r_{rb}$).

## Results

There were no significant differences in demographic variables and prior CPR knowledge between the UBV and LFV conditions ($p > 0.05$ for all).

For all QCPR metrics excluding CCF and average compressions per minute, there were no statistically significant differences found between groups ($p > 0.05$). With regard to CCF percentage scores, the UBV group (Mdn = 98, IQR = 7.25) had a significantly higher score than the LFV group (Mdn = 89, IQR = 15), and a large effect size was observed ($r_{rb}$ = -0.37, $Z$ = -2.79, adjusted $p$ = 0.02).The LFV group had significantly higher average compressions per minute (Mdn = 117 bpm, IQR = 4 bpm) than the UBV group (Mdn = 112 bpm, IQR = 17.5 bpm; $Z$ = -3.17, adjusted $p$ = 0.01) and a large effect size was observed ($r_{rb}$ = 0.42). However, both of these compression rates fell within the clinically recommended range of 100–120 compressions per minute [31]. Feedback from Scottish Ambulance Service paramedics and call handlers on the simulation video data confirmed that the MPDS script was being used adequately and that the simulations were being performed accurately. A summary of the QCPR performance metrics for each group can be found in Table 2.

Regarding the variables relevant to shared social identity, video intervention feedback, and group attitudes towards performing CPR, none of the survey metrics aside from shared social identity with the video instructor and expected support were found to have statistically significant inter-group differences ($p > 0.05$ for all). After adjusting for multiple comparisons, only mean differences in expected support remained significant. Participants in the UBV condition (Median = 9, IQR = 2) had greater feelings of expected support from Scottish people than the

**Table 2. Descriptive statistics for the QCPR metrics observed in each condition.**

| Variable | UBV (n = 38) | | LFV (n = 37) | |
|---|---|---|---|---|
| | **Median** | **IQR** | **Median** | **IQR** |
| **CCF * (%)** | 98 | 7.25 | 89 | 15 |
| **Compression Depth (%)** | 86 | 76.25 | 96 | 37 |
| **Release (%)** | 100 | 3.5 | 100 | 1 |
| **Overall Compression Score (%)** | 91 | 37.5 | 91 | 15 |
| **Total Compressions** | 214 | 54.75 | 202 | 37 |
| **Compressions between 100–120 bpm (%)** | 58 | 43 | 59 | 35 |
| **Average Compressions per Minute *** | 112 | 17.5 | 117 | 4 |
| **Total CPR Score (%)** | 68 | 66.25 | 56 | 74 |

*A summary of the QCPR variables per condition. Medians and interquartile ranges (IQR) are presented.*

Note

* = statistically significant differences after adjusting for multiple comparisons.

LFV group (*Median* = 8, *IQR* = 2, Z = -2.86, *p* = 0.048) with a large effect size ($r_{rb}$ = -0.35) observed. Importantly, there were no significant group differences in terms of attitudes towards performing CPR or perceived clarity and understanding of the respective video instructions. A summary of the post-intervention survey metrics for each group can be found in Table 3.

## Discussion

This study demonstrated that aside from the variables of CCF (where the UBV condition demonstrated significantly higher scores) and average compressions per minute (where the LFV condition had significantly higher scores, yet both condition's median scores fell within the clinically acceptable range of 100–120 bpm), there were no significant differences in CPR performance between participants who watched a 2-minute UBV CPR intervention in

**Table 3. Descriptive statistics for the social identity and video feedback variables across conditions.**

| Variable | UBV (*n* = 42) | | LFV (*n* = 40) | | |
|---|---|---|---|---|---|
| | **Median** | **IQR** | **Median** | **IQR** | **Max. Possible Score** |
| **Scottish Social Identity** | 14 | 5 | 14 | 5.5 | 20 |
| **Shared Social Identity with the Video Instructor** | 12 | 3.75 | 11 | 3 | 15 |
| **Trust in the Person Providing Instruction** | 5 | 1 | 5 | 0 | 5 |
| **Perceived Norm of Helping** | 4 | 1 | 4 | 1 | 5 |
| **Expected Support *** | 9 | 2 | 8 | 2 | 10 |
| **Confidence in Performing CPR** | 12 | 3 | 12 | 2 | 15 |
| **Willingness to Perform CPR** | 8 | 1 | 8.5 | 2 | 10 |
| **Anxiety from the Video Instructions** | 9 | 2 | 9 | 2 | 10 |
| **Sufficiency of Practical CPR Information** | 14 | 3 | 15 | 2 | 15 |
| **Clarity** | 15 | 1 | 15 | 2.25 | 15 |
| **Barriers to CPR Performance** | 11 | 3.75 | 10.5 | 5.25 | 20 |

Summary of the social identity and video feedback variables. Medians, interquartile ranges (IQR), and maximum possible scores are presented. High scores indicate a stronger presence of the variable of interest.

Note

* = statistically significant differences after adjusting for multiple comparisons.

conjunction with a SALFS bag manikin and those who watched a 16-minute LFV intervention with an incorporated Laerdal Mini Anne manikin. Most notably, there were no statistically significant differences between conditions on the overall CPR composite score. Regarding the social identity variables, participants in the UBV condition reported significantly higher levels of expected support from other Scottish people in an emergency when compared to the LFV group. There were no differences between conditions in relation to attitudes towards performing CPR. Both groups felt their respective videos provided sufficient CPR information and clarity of instruction, and there were no significant differences in perceived barriers to CPR and anxiety after watching the video instructions.

Previous research has demonstrated the efficacy of UBVs as CPR interventions. Bobrow and colleagues [14] found that participants exposed to UBVs had significantly higher average compression rate and depth scores than an untrained control group. Similarly, Panchal and colleagues [10] found that participants who viewed a UBV were more likely to call for emergency services and faster to begin chest compressions than an untrained control group. While these studies demonstrate the utility of UBV CPR familiarisation in comparison to untrained laypersons, the current study expands upon these findings by evaluating the effectiveness of a UBV in comparison to a LFV intervention in the context of emergency services-led bystander CPR. By demonstrating no clinically significant differences between the UBV and LFV conditions (excluding the variable of CCF, in which the UBV condition scored higher than the LFV condition), this study provides an empirical basis for the efficacy of UBVs in the context of CPR familiarisation.

The findings of this study hold key implications for both CPR familiarisation and wider public health training initiatives. UBV familiarisation could be a key method to equip traditionally hard-to-reach groups [9,12] and those most at risk of witnessing a cardiac arrest, who often find it difficult to access or engage with traditional methods of instruction [12,15,32]. Furthermore, UBV interventions could be used to supplement traditional CPR training methods as a means of refreshing knowledge and enabling easy sharing of CPR skills, for example on social media. This UBV is also unique in that it considers the wider system of resuscitation —explaining CPR in a realistic context that includes the emergency services call handler, the patient, and the bystander.

A strong body of research evidences the role of social identity processes in emergency helping behaviours [19,21,22]. Research conducted by Drury and colleagues found that bystanders are more likely to provide support to and expect support from members of their own social group in an emergency context [22,33]. Similarly, past research has found that framing helping behaviour as being intrinsic to particular social identities resulted in increased helping and health-related behaviours in the associated social group [17,20,34]. The current study considered multiple avenues through which social identity could be harnessed in the UBV to facilitate helping behaviours. These included raising the salience of Scottish social identity (e.g., casting an actor with a Scottish accent), while also framing the bystander, instructor, emergency services, and the patient as being part of the same social group. The UBV script explicitly defined social norms associated with Scottish identity (e.g., including lines such as "in Scotland we look out for each other"). Participants who watched the UBV demonstrated significantly higher questionnaire scores in 'expected support from other Scottish people' compared to the non-Scottish LFV. This finding holds key implications for the use of social identity priming in CPR training materials, as people have been shown to adhere to behaviours that they believe are expected of them [20,35,36]. Furthermore, these results highlight the potential for using social identity priming as a means of facilitating bystander compliance with emergency instructions, as people are more likely to follow emergency instructions from those seen as part of their social group [21,37,38]. This social identification with others in an emergency

relates to collective agency, or working with others to overcome an obstacle, and a willingness to help others [38]. These findings suggest that increasing the salience of national identity and presenting CPR as a social norm is possible in the context of even a UBV, and that social identity priming may be a potential mechanism for encouraging bystander CPR.

This study also presented some novel methodological approaches for evaluating CPR interventions. The research team developed a portable simulation suite in collaboration with Motorola Solutions which enabled the running of simulations in a more naturalistic environment, as opposed to being in a stationary, clinical simulation suite. This maintained greater ecological validity and allowed richer data collection on CPR quality using multi-angle video recordings of the simulation, which enabled feedback from Scottish Ambulance Service call handlers and paramedics on simulation accuracy. Fidelity was also maximised by including a trained emergency services call handler in the simulation, a component often omitted in OHCA simulation research. In the majority of OHCA cases, bystander resuscitation is facilitated by an emergency services call handler [39–41]. The addition of a call handler in the current study enabled testing of the adequacy of UBV CPR familiarisation in the context of a real-world system of care.

Finally, there are key study limitations to be considered. This study took place during the COVID-19 lockdowns, with government restrictions making recruitment of larger samples drawn from outside the University of Edinburgh impossible. While we endeavoured to recruit a diverse and varied sample within the university, this sample is not representative of the general public [42]. Therefore, the findings of this study cannot be generalised beyond a university setting to the public at large. Future work could examine the efficacy of socially primed UBVs using a more representative sample for the purposes of generating more generalisable results. Also, this study did not examine decay of CPR performance quality over time [43]. Future research could explore post-UBV CPR performance longitudinally to further probe its long-term efficacy and the need to refresh CPR knowledge.

## Conclusions

To the best of the authors' knowledge, this study is the first to evaluate the effectiveness of a low-cost, shareable UBV CPR intervention in comparison to a traditional LFV intervention. Furthermore, it is the first study to employ a social identity approach to CPR UBV intervention development with the goal of increasing the likelihood of bystander CPR. These findings suggest that UBV interventions are a viable way to equip the public with basic CPR knowledge, and that priming the viewer with collectivist language and increasing the salience of a shared sense of social identity impacts the level of expected support from others. This method of CPR knowledge dissemination holds promise for maximising the shareability of CPR skills, and for potentially empowering bystander action through social identity priming.

## Supporting information

**S1 Appendix. Summary of UBV script development and testing.**
(DOCX)

**S2 Appendix. Laerdal QCPR metrics.**
(DOCX)

## Acknowledgments

The authors would like to thank the following people for their invaluable assistance during the course of this project: Donald McPhail and Judith Richardson of the Scottish Ambulance

Service; Eilish Murphy, James Nicholson, Dominika Skrocka, Susan Gardner, Diane Lac, David Souter, Brian Gilhooley, Elsbeth Dewhirst, Sara Illicic and Molly Brewster of the University of Edinburgh/NHS Lothian; and all the volunteers who took part.

## Author Contributions

**Conceptualization:** Jean Skelton, Anne Templeton, Jennifer Dang Guay, Lisa MacInnes, Gareth Clegg.

**Data curation:** Jean Skelton.

**Formal analysis:** Jean Skelton.

**Funding acquisition:** Jean Skelton, Anne Templeton, Lisa MacInnes, Gareth Clegg.

**Investigation:** Jean Skelton.

**Methodology:** Jean Skelton, Anne Templeton, Jennifer Dang Guay, Gareth Clegg.

**Project administration:** Jean Skelton, Jennifer Dang Guay.

**Resources:** Anne Templeton, Jennifer Dang Guay.

**Supervision:** Jean Skelton, Anne Templeton, Lisa MacInnes, Gareth Clegg.

**Validation:** Anne Templeton.

**Visualization:** Jean Skelton.

**Writing – original draft:** Jean Skelton.

**Writing – review & editing:** Jean Skelton, Anne Templeton, Jennifer Dang Guay, Lisa MacInnes, Gareth Clegg.

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
