## [Decision Letter · Decision Letter 0]

2 May 2024

PONE-D-24-00549Developing and evaluating a brief, socially primed video intervention to enable bystander cardiopulmonary resuscitation: A randomised control trialPLOS ONE

Dear Dr. Skelton,

Thank you for submitting your manuscript to PLOS ONE. After careful consideration, we feel that it has merit but does not fully meet PLOS ONE’s publication criteria as it currently stands. Therefore, we invite you to submit a revised version of the manuscript that addresses the points raised during the review process.

We look forward to receiving your revised manuscript.

Kind regards,

David Wampler

Academic Editor

PLOS ONE

Journal Requirements:

Additional Editor Comments:

Dear Dr. Skelton

I would like to thank you for considering PLOS ONE for the publication of your manuscript “Developing and evaluating a brief, socially primed video intervention to enable bystander cardiopulmonary resuscitation: A randomised control trial”.

I also very much appreciate your patience during the review process.

Our expert reviewers have now considered your work and the reviews were quite favorable, and I am confident that with only minor revision we will be able to move forward with making a final decision. I have included an edited version of reviewer comments below, and have added a few comments from an editor perspective.

Thank you again and I look forward to your careful revision,

David Wampler,

Editor.

Review of PONE-D-24-00549

Overall: Important topic, addresses key components of behavioral psychology and cultural anthropology concerns as barriers to critical clinical intervention; joins “wagging the dog” and “assuming the sale” as a way to mitigate bystander effect/unwillingness to act. Builds on previous work in the field(s), and should prove replicable in analogous cultural contexts. Most statistical methodology appears appropriate vs the data. Discussion and conclusion are measured and sufficiently supported by results. Statistical methodology for script development well outlined in supplemental materials.

Major issues: None noted.

Minor issues:

Manuscript page 10, line 57: hyphenate information-heavy?

Manuscript page 12, line 105-110: Rationale for different training equipment (manikin v bag) understandable, but ideally as few variables as possible would have been changed – granted this is effectively an intention-to-treat paradigm, wherein the entire training model is being compared to another, rather than only the video component. Hoping this is discussed later. (edit: I don’t believe it was mentioned)

Results section of the abstract: please rework this just a bit, the current version lacks the detail presented in the main body of the manuscript.

Results of the main manuscript: authors state that eight subjects were excluded due to technical issues. Please report from how many were excluded from each study arm.

Table 1: Worth noting in the table that the ages, though visibly skewed older in the LFV group, does not meet statistically significant difference (at least according to my napkin-math Fisher’s Exact), nor does the employment/student category. The first impulse I had upon looking at it was that the younger cohort got the short-attention-span friendly video – but speaking to nonsignificant difference may allay some of that objection.

Manuscript page 15, line 298: Something of a quibble, but when using nonparametric tests for everything else, working in Cohen’s d (a parametric test calculated using means and SDs) seems incongruous. While only the exceptionally nerdy (e.g., me) would notice or care – res ipsa loquitur for most of the findings – I’d be remiss not to mention it. Rank biserial is the nonpar equivalent, if wishing to modify. Or dismiss out of hand because the end state is almost certainly the same.

Tables 2,3: There are a lot of tests being run here, and no mention of multiple comparisons adjustments against minimum three hypotheses, and at least 19 tests identifiable amongst the tables. Granted, the consequences of type I error in this scenario are profoundly low – just something to consider regarding internal consistency.

Figure two is too low in resolution to be publication quality, please provide a better version.

Reviewers' comments:

Reviewer's Responses to Questions

**Comments to the Author**

1. Is the manuscript technically sound, and do the data support the conclusions?

Reviewer #1: Yes

2. Has the statistical analysis been performed appropriately and rigorously? 

Reviewer #1: Yes

3. Have the authors made all data underlying the findings in their manuscript fully available?

Reviewer #1: Yes

4. Is the manuscript presented in an intelligible fashion and written in standard English?

Reviewer #1: Yes

5. Review Comments to the Author

Reviewer #1: Review of PONE-D-24-00549

Overall: Important topic, addresses key components of behavioral psychology and cultural anthropology concerns as barriers to critical clinical intervention; joins “wagging the dog” and “assuming the sale” as a way to mitigate bystander effect/unwillingness to act. Builds on previous work in the field(s), and should prove replicable in analogous cultural contexts. Most statistical methodology appears appropriate vs the data. Discussion and conclusion are measured and sufficiently supported by results. Statistical methodology for script development well outlined in supplemental materials.

Major issues: None noted.

Minor issues:

Manuscript page 10, line 57: hyphenate information-heavy?

Manuscript page 12, line 105-110: Rationale for different training equipment (manikin v bag) understandable, but ideally as few variables as possible would have been changed – granted this is effectively an intention-to-treat paradigm, wherein the entire training model is being compared to another, rather than only the video component. Hoping this is discussed later. (edit: I don’t believe it was mentioned)

Table 1: Worth noting in the table that the ages, though visibly skewed older in the LFV group, does not meet statistically significant difference (at least according to my napkin-math Fisher’s Exact), nor does the employment/student category. The first impulse I had upon looking at it was that the younger cohort got the short-attention-span friendly video – but speaking to nonsignificant difference may allay some of that objection.

Manuscript page 15, line 298: Something of a quibble, but when using nonparametric tests for everything else, working in Cohen’s d (a parametric test calculated using means and SDs) seems incongruous. While only the exceptionally nerdy (e.g., me) would notice or care – res ipsa loquitur for most of the findings – I’d be remiss not to mention it. Rank biserial is the nonpar equivalent, if wishing to modify. Or dismiss out of hand because the end state is almost certainly the same.

Tables 2,3: There are a lot of tests being run here, and no mention of multiple comparisons adjustments against minimum three hypotheses, and at least 19 tests identifiable amongst the tables. Granted, the consequences of type I error in this scenario are profoundly low – just something to consider regarding internal consistency.

6. PLOS authors have the option to publish the peer review history of their article (what does this mean?). If published, this will include your full peer review and any attached files.

Reviewer #1: **Yes: **Ian L Hudson, DO, MPH

---

## [Author Response · Author response to Decision Letter 0]

8 May 2024

Dear Dr Wampler,

Thank you for giving us the opportunity to resubmit a revised copy of the manuscript “Developing and evaluating a brief, socially primed video intervention to enable bystander cardiopulmonary resuscitation: A randomised control trial” for PLOS ONE. We appreciate the valuable feedback you and the reviewer provided on the paper. We have incorporated these improvements into the current draft of the manuscript, which are highlighted with Track Changes, Please see below our responses to each feedback point. Please note that line numbers correspond to the Track Changes manuscript draft:

Reviews’ comments to the authors:

1. Manuscript page 10, line 57: hyphenate information-heavy?

Author response: Thank you for pointing this out. “Information-heavy” has been hyphenated for clarity and grammatical correctness (Line 88).

2. Manuscript page 12, line 105-110: Rationale for different training equipment (manikin v bag) understandable, but ideally as few variables as possible would have been changed – granted this is effectively an intention-to-treat paradigm, wherein the entire training model is being compared to another, rather than only the video component. Hoping this is discussed later. (edit: I don’t believe it was mentioned)

Author response: We agree with the reviewer that the rationale for using a bag manikin in the ultra-brief condition could be more explicitly explained in the text. As the reviewer correctly points out, the intervention being tested was the training model in its entirety (i.e. both the manikin and the video as a unit), rather than just the familiarisation videos in isolation. As accessibility to CPR familiarisation is a core tenet of our ultra-brief intervention, and compression practice using a manikin is a key quality indicator of resulting CPR skills, we decided to use the freely availably/printable bag manikin for our ultra-brief condition. This bag manikin fit more with the ethos of this intervention, as inflatable Mini Anne manikins usually come with a cost. We have added a passage to the background section to highlight this rationale (Lines 134, 142-144; 167-168).

3. Results section of the abstract: please rework this just a bit, the current version lacks the detail presented in the main body of the manuscript. 

Author response: Thank you for this feedback, we have reworked the abstract to provide a more detailed summary of the results section that is more in line with the results presented in the main text. (Lines 21-50)

4. Results of the main manuscript: authors state that eight subjects were excluded due to technical issues. Please report from how many were excluded from each study arm.

Author response: We agree that the data lost to technical issues could be more clearly stated. We have specified the number of participants excluded from the QCRP dataset for each condition due to technical issues with our QCPR manikin. Prior to submitting the manuscript for publication, data from one participant that had been lost to a technical issue was recovered. This passage was not updated to reflect this, and so has now been updated with the correct details on numbers of participants excluded per study arm (Lines 228-230).

5. Table 1: Worth noting in the table that the ages, though visibly skewed older in the LFV group, does not meet statistically significant difference (at least according to my napkin-math Fisher’s Exact), nor does the employment/student category. The first impulse I had upon looking at it was that the younger cohort got the short-attention-span friendly video – but speaking to nonsignificant difference may allay some of that objection.

Author response: The reviewer raises a good point here – we conducted Fisher’s Exact Tests on all demographic variables to ensure there were no significant inter-group differences between our conditions. We have highlighted this more clearly in Table 1. (Lines 225-226)

6. Manuscript page 15, line 298: Something of a quibble, but when using nonparametric tests for everything else, working in Cohen’s d (a parametric test calculated using means and SDs) seems incongruous. While only the exceptionally nerdy (e.g., me) would notice or care – res ipsa loquitur for most of the findings – I’d be remiss not to mention it. Rank biserial is the nonpar equivalent, if wishing to modify. Or dismiss out of hand because the end state is almost certainly the same.

Author response: Thank you for this informative feedback – the reviewer is correct that a non-parametric effect size would make more sense in the context of our analysis. While the reviewer is correct that this did not change the end state, we want to be consistent with our analysis, and so have replaced our Cohen’s d values with rank-biserial values (Lines 335-336; 344-345; 348; 366-367).

7. Tables 2,3: There are a lot of tests being run here, and no mention of multiple comparisons adjustments against minimum three hypotheses, and at least 19 tests identifiable amongst the tables. Granted, the consequences of type I error in this scenario are profoundly low – just something to consider regarding internal consistency.

Author response: The reviewer raises an important point here regarding multiple comparisons and Type I Error. When planning our analysis, this was something we considered carefully. As we were working with 2 separate datasets (the QCPR participants and the post-intervention survey datasets), we felt that our analyses were adequately robust against Type I Error without additional corrections for multiple comparisons. We also considered that controlling for multiple comparisons in this case could result in an overly conservative thresholds for significance that may end up masking meaningful findings. However, we recognise that the risk of Type I Error in this case is not zero, and therefore have added Benjamini-Hochberg-corrected p-values to the Results section to be fully rigorous in our analyses. This method was chosen as it is suitable for controlling the False Discovery Rate when conducting multiple comparisons with non-parametric tests, while preserving statistical power (Lines 332-333; 344-345; 348; 356; 360-364; 366-367; 372; 373; 384-386; 429-431; 470-471).

8. Figure two is too low in resolution to be publication quality, please provide a better version.

Author response: We apologise for this low resolution figure; we have attached a higher resolution 300 DPI version of our Figure 2 that we hope will suffice for publication quality.

---

## [Decision Letter · Decision Letter 1]

7 Jun 2024

PONE-D-24-00549R1Developing and evaluating a brief, socially primed video intervention to enable bystander cardiopulmonary resuscitation: A randomised control trialPLOS ONE

Dear Dr. Skelton,

Thank you for submitting your manuscript to PLOS ONE. After careful consideration, we feel that it has merit but does not fully meet PLOS ONE’s publication criteria as it currently stands. Therefore, we invite you to submit a revised version of the manuscript that addresses the points raised during the review process.

We look forward to receiving your revised manuscript.

Kind regards,

David Wampler

Academic Editor

PLOS ONE

Journal Requirements:

Additional Editor Comments:

Dear Dr. Skelton.

Congratulations on I received outstanding reviews on your paper "Developing and evaluating a brief, socially primed video intervention to enable bystander cardiopulmonary resuscitation: A randomised control trial".

The only request is to add a comment in the limitations section that identifies that the population included were limited to those affiliated with an academic organization, and this may not be representative of society in general.

Below are an edited version of reviewer comments:

Overall: Fascinating blend of behavioral conditioning/propaganda used for good and rigorously defensible methodology. All reviewer observations thoroughly addressed. It is hoped, as a result of such research, that organizations like the American Red Cross and American Heart Association abandon four hour classes in favor of something more efficient; the British appear to be forward-leaning with the concept the authors promulgate – RevivR program currently front-facing on their website.

Major issues: None noted.

Minor issues:

Manuscript page 11, line 216: The participants are not random people, but University students and staff. While the limitations section notes the COVID-related challenge, it does not currently examine problems with extrapolating the findings to the lay public at large, at the moment one can only say that amongst those already selected for intelligence and reliability, UBV appears to serve the requirement.

Reviewers' comments:

Reviewer's Responses to Questions

**Comments to the Author**

1. If the authors have adequately addressed your comments raised in a previous round of review and you feel that this manuscript is now acceptable for publication, you may indicate that here to bypass the “Comments to the Author” section, enter your conflict of interest statement in the “Confidential to Editor” section, and submit your "Accept" recommendation.

Reviewer #1: All comments have been addressed

2. Is the manuscript technically sound, and do the data support the conclusions?

Reviewer #1: Yes

3. Has the statistical analysis been performed appropriately and rigorously? 

Reviewer #1: Yes

4. Have the authors made all data underlying the findings in their manuscript fully available?

Reviewer #1: Yes

5. Is the manuscript presented in an intelligible fashion and written in standard English?

Reviewer #1: Yes

6. Review Comments to the Author

Reviewer #1: Overall: Fascinating blend of behavioral conditioning/propaganda used for good and rigorously defensible methodology. All reviewer observations thoroughly addressed. It is hoped, as a result of such research, that organizations like the American Red Cross and American Heart Association abandon four hour classes in favor of something more efficient; the British appear to be forward-leaning with the concept the authors promulgate – RevivR program currently front-facing on their website.

Major issues: None noted.

Minor issues:

Manuscript page 11, line 216: Embarrassed at not noticing this the first time around – the participants are not random people, but University students and staff. While the limitations section notes the COVID-related challenge, it does not currently examine problems with extrapolating the findings to the lay public at large. It serves as a fine proof of concept, and needs little more than a head-nod to the issue, but at the moment one can only say that amongst those already selected for intelligence and reliability, UBV appears to serve the requirement.

Manuscript page 21, line 392, and line 395: In the spirit of ongoing pedantry on this reviewer’s part, the authors are asked to consider that “i.e.” (id est – that is to say – a more specific paraphrase or revealed implication) be replaced with “e.g.” (exempli gratia – free example/for the sake of example, ideally out of a greater unstated list) in these cases. I.e. is used correctly (in this reviewer’s estimation) higher up in manuscript page 5, line 88, while in manuscript page 5, line 96, the problem is avoided by spelling out “such as.” I realize I could have said nothing and the outcome would be unchanged; observations like this are not a hit at parties. Overall, the language of the manuscript is a pleasure to read.

General comment: If I worked for the Scottish Ambulance Service, I would absolutely tell people I worked for the SAS and refuse to elaborate, claiming it was classified.

7. PLOS authors have the option to publish the peer review history of their article (what does this mean?). If published, this will include your full peer review and any attached files.

Reviewer #1: **Yes: **Ian L Hudson, DO, MPH

---

## [Author Response · Author response to Decision Letter 1]

7 Jun 2024

Dear Dr Wampler,

Thank you for giving us the opportunity to resubmit a revised copy of the manuscript “Developing and evaluating a brief, socially primed video intervention to enable bystander cardiopulmonary resuscitation: A randomised control trial” for PLOS ONE. We appreciate the valuable feedback you and the reviewer provided on the paper. We have incorporated these improvements into the current draft of the manuscript, which are highlighted with Track Changes, Please see below our responses to each feedback point in blue font. Please note that line numbers correspond to the Track Changes manuscript draft:

Reviews’ comments to the authors:

1. Manuscript page 11, line 216: Embarrassed at not noticing this the first time around – the participants are not random people, but University students and staff. While the limitations section notes the COVID-related challenge, it does not currently examine problems with extrapolating the findings to the lay public at large. It serves as a fine proof of concept, and needs little more than a head-nod to the issue, but at the moment one can only say that amongst those already selected for intelligence and reliability, UBV appears to serve the requirement.

Author response: Thank you for pointing this out. We have added a passage to the limitations section that more explicitly outlines that the study employed a university sample and that therefore results cannot be generalised to the wider public. We tied this into our recommendation that future research explore the utility of UBV CPR interventions with a more representative sample for the purposes of generating more generalisable findings (lines 425-430).

2. Manuscript page 21, line 392, and line 395: In the spirit of ongoing pedantry on this reviewer’s part, the authors are asked to consider that “i.e.” (id est – that is to say – a more specific paraphrase or revealed implication) be replaced with “e.g.” (exempli gratia – free example/for the sake of example, ideally out of a greater unstated list) in these cases. I.e. is used correctly (in this reviewer’s estimation) higher up in manuscript page 5, line 88, while in manuscript page 5, line 96, the problem is avoided by spelling out “such as.” I realize I could have said nothing and the outcome would be unchanged; observations like this are not a hit at parties. Overall, the language of the manuscript is a pleasure to read.

Author response: On the contrary, we greatly appreciate the reviewer’s attention to detail! “i.e.” has been changed to “e.g.” in both cases for grammatical correctness (lines 392; 395).

3. General comment: If I worked for the Scottish Ambulance Service, I would absolutely tell people I worked for the SAS and refuse to elaborate, claiming it was classified.

Author response: The corresponding author now regrets not taking up a contract with the ambulance service.

---

## [Editor Report · Decision Letter 2]

11 Jun 2024

Developing and evaluating a brief, socially primed video intervention to enable bystander cardiopulmonary resuscitation: A randomised control trial

PONE-D-24-00549R2

Dear Dr. Skelton,

We’re pleased to inform you that your manuscript has been judged scientifically suitable for publication and will be formally accepted for publication once it meets all outstanding technical requirements.

Kind regards,

David Wampler

Academic Editor

PLOS ONE

Additional Editor Comments (optional):

Dear Dr. Skelton,

Thank you for your patients as we have worked to push this paper to publication. Congratulations, I have now recommended this manuscript for acceptance.

This is not the final step, there is still some work to do, but congratulations on a job well done.

Sincerely,

David Wampler, PhD, LP, FAEMS

Editor
---

## [Editor Report · Acceptance letter]

25 Jun 2024

PONE-D-24-00549R2 

PLOS ONE

Dear Dr. Skelton, 

I'm pleased to inform you that your manuscript has been deemed suitable for publication in PLOS ONE. Congratulations! Your manuscript is now being handed over to our production team.

Kind regards, 

on behalf of

Dr. David Wampler 

Academic Editor

PLOS ONE